# SIMILARITY GROUPS SCALE FAIRNESS WITHOUT DEMOGRAPHIC DATA

## ABSTRACT

Intelligent agent systems increasingly mediate interactions between humans and digital environments, making ethical guidance a critical component of their development. To achieve robust models capable of interaction with human, Self-Supervised Learning (SSL) has emerged as a dominant paradigm. This approach enables foundation models to be built only with the raw data, supporting reliable performance in real-world environments. However, ensuring fairness in these systems remains a major challenge, as existing methods for evaluation depend heavily on explicit metadata which are difficult to obtain at scale. Here, we introduce a unified framework for Similarity Group Fairness (SGF), a synthesis of group and individual fairness notions, that enables the auditing and safeguards of fairness without relying on demographic metadata.

## 1 INTRODUCTION

The rapid advancement of artificial intelligence has prompted a shift from isolated models to autonomous agents operating within complex socio-economic ecosystems Fang et al. (2025); Carichon et al. (2025). As these systems increasingly mediate high-stakes decisions, such as those regulated by the EU AI Act in healthcare, ethical alignment becomes a technical imperative. Self-supervised learning (SSL) has emerged as a dominant paradigm for developing foundation models that demonstrate enhanced robustness and inherent fairness in computer vision Goyal et al. (2022), even without extensive data curation. Nonetheless, systematic methods for evaluating and optimizing these models remain necessary to ensure equitable interactions between autonomous agents and society.

Fairness in algorithmic systems is commonly framed through two principles: group fairness Barocas et al. (2023) and individual fairness Dwork et al. (2011). Group fairness seeks statistical parity across demographic groups, while individual fairness requires that similar individuals receive comparable outcomes. These concepts underpin data curation, model training, evaluation, and policy design Queiroz et al. (2025b). Although group fairness is more widely adopted due to its computational feasibility Fleisher (2021), compliance at the group level does not necessarily imply equity at the individual level Dwork et al. (2011).

Despite their significance, demographic attributes essential for fairness assessment are often costly to collect and restricted by privacy regulations, particularly in healthcare Queiroz et al. (2025b). This constraint creates a methodological paradox, as fairness evaluation depends on metadata that is frequently unavailable. Moreover, when such data exist, they often impose simplistic categorizations such as binary definitions or arbitrary divisions of age that fail to capture the intersectional complexity of human identities Chen et al. (2023). A more nuanced representation of diversity is therefore necessary to advance fair models.

## 2 SIMILARITY GROUP FAIRNESS

Recent work on slice discovery shows that fairness can be assessed without demographic labels Eyuboglu et al. (2022); Queiroz et al. (2025a). Subgroup discovery clustering methods reveal fairness gaps that surpass those identified through demographic-based analyses Bissoto et al. (2025). Building on this insight, we introduce Similarity Groups (SGs), which unify individual and group fairness by defining groups through representation similarity rather than discrete attributes. This for-

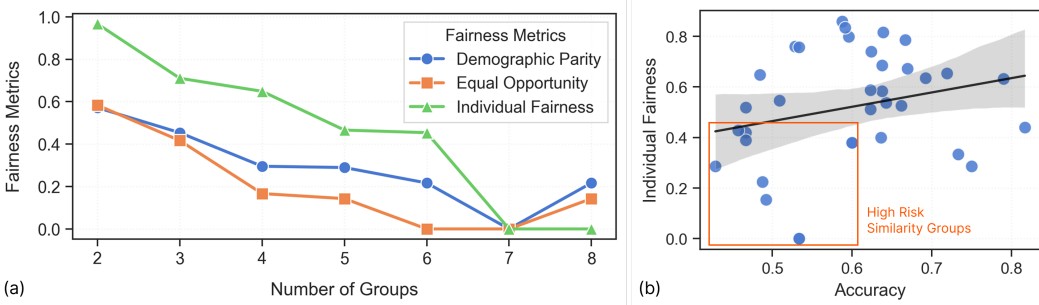

Figure 1: (a) Demographic Parity, Equal Opportunity, and Individual fairness are each reported as the ratio of the smallest to the largest group-level rate across Similarity Groups. For Individual fairness, we compute per SGs the percentage of pairs that do not violate Equation 2; the figure reports the ratio of the smallest to the largest of these percentages. A ratio of 1 indicates equal rates across groups. (b) Correlation between individual fairness and accuracy across all SGs from 2 to 8 groups.

mulation provides a scalable and robust auditing framework for fairness in the absence of sensitive human annotations. We formally define SGs as follows:

**Definition 1** (Similarity Group). *Let $\mathcal{X}$ be a population of individuals and $\phi : \mathcal{X} \to \mathbb{R}^d$ be an embedding map that represents each individual $x \in \mathcal{X}$ in a $d$-dimensional latent space. Given a distance metric $d : \mathbb{R}^d \times \mathbb{R}^d \to \mathbb{R}_{\geq 0}$ in the embedding space and a set of centroids $z_1, \ldots, z_N \subset \mathbb{R}^d$ in the embedding space, a similarity group $SG_i \subseteq \mathcal{X}$ is defined as:*

$$SG_i = \big\{ x \in \mathcal{X} \mid d\big(\phi(x), z_i\big) \leq d\big(\phi(x), z_j\big), \ \forall j \in \{1, \ldots, N\}\big\}. \tag{1}$$

The framework achieves statistical parity, a core group fairness criterion, by enforcing equal outcomes across SG-defined subgroups. For individual fairness, we adopt the $L$-Lipschitz condition on a classifier $h : X \to Y$ following Dwork et al. (2011), where $X$ denotes the input space, $Y$ denotes the classifier output space (e.g., softmax-normalized logits), and extend it to the embedding space defined by $\phi$ over pairs of individuals within each SG, formalized as:

$$d_Y(h(x_1), h(x_2)) \leq L d_X(\phi(x_1), \phi(x_2)) \quad \forall x_1, x_2 \in X, \tag{2}$$

where $d_X$ denotes the distance between embeddings $\phi(x_1)$ and $\phi(x_2)$, and $d_Y$ the distance between classifier output space. Varying the number of SGs $N$ controls the granularity at which these group and individual notions are evaluated.

**Experimental Setup.** To evaluate SGs empirically, experiments employ the tuple $(\mathcal{X}, \phi, h)$, where $\mathcal{X}$ denotes the dataset, $\phi$ the embedding model, and $h$ the classifier. For the medical imaging case, the CheXpert Irvin et al. (2019) test set supplies $\mathcal{X}$ with chest X-ray images to assess $h$, implemented as a ConvNeXt model trained for multi-label prediction across five classes. Groups derive from $\phi$ via RadDINO Pérez-García et al. (2025) embeddings, clustered using hierarchical k-means Vo et al. (2024) after PCA dimensionality reduction to 2D, preserving distance relationships while enhancing computational efficiency. Embeddings are L2-normalized, logits are softmax-transformed, and distances are computed using the L2 Euclidean distances.

**Results and discussion.** Figure 1(a) shows that embedding-defined subgroups exhibit implicit model disparities, consistent with prior findings Bissoto et al. (2025). The number of SGs clusters $N$ controls the evaluation granularity over $\mathcal{X}$: $N = 2$ yields general representations, while higher values of $N$ reveal specific risk groups. For $N = 7$, fairness ratios reach zero, thereby identifying perfectly high risk SGs. Figure 1(b) shows a positive correlation between individual fairness and accuracy: subgroups with higher individual fairness exhibit higher accuracy. This relationship enables the use of embeddings $\phi(x)$ for $x \in \mathcal{X}$ to identify instances belonging to subgroups with both low accuracy and low individual fairness; such identification can serve as a safeguard in deployment by withholding model predictions for those instances. The framework scales fairness evaluation in the wild using only population $\mathcal{X}$ data. Although embedding-based representations entail known limitations Weller et al. (2025), the approach nevertheless enables scalable fairness auditing without demographic metadata and may be strengthened by future advances in representation learning.

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
