# OpenReview forum: "Similarity Groups Scale Fairness Without Demographic Data"
_ICLR.cc/2026/Workshop/AFAA — Submitted to AFAA 2026_

### Official Review · Reviewer_vqL2 · 2026-02-21
**Review for "Similarity Groups Scale Fairness Without Demographic Data"**

**Rating:** 3
**Confidence:** 3

**Summary:**

In this paper, the authors consider the task of training (to induce) or measuring  (to evaluate) a model's individual and/or subgroup fairness in settings where the sociodemographic attributes required to assess similarity among individuals, or infer subgroup membership are unavailable or noisy. To address this limitation, the authors introduce a framework, Similarity Group Fairness (SGF). The main idea here is that individuals can be represented as real-valued embeddings, and (fairness-relevant) groups can be defined by clustering and considering the set of individuals who share a cluster centroid. The authors posit that this framing makes it possible to (eg) consider/enforce statistical parity with respect to cluster-defined subgroups, and imposing a Lipschitz-style individual fairness constraint.

**Strengths:**

The problem that the authors consider is well-motivated: it is common not only in the healthcare settings that they focus on, but in many real-world socially consequential settings, and is particularly problematic when the demographic information (or other fairness-relevant metadata) may be missing-not-at-random. Similar challenges also arise in other settings, such as recommendation/e-commerce systems, where (for privacy purposes) an individual's sensitive attributes may not be available, but the system designer may still wish to intervene in the system to evaluate / improve fairness for some subset of market participants, and embeddings about (eg) the content with which they interact may be readily available. The empirical results presented are early-stage, but do suggest the approach can be useful for identifying at-risk subgroups in latent space.

**Weaknesses:**

The primary weakness I perceive that (if addressed) will help to strengthen the points the authors make, is that the success/utility of the framework they introduce (wrt improving / evaluating fairness-related claims) relies heavily on the quality of the embedding map used, and many common choices for specific modalities have been demonstrated to contain inductive priors which encode/amplify biases (against people/groups) that exist in the real-world images/text these models have been trained on. This could (eg) result in issues due to latent confounders ~ eg , in a setting where unstructured clinical text is used instead of x-ray images, patients who are similar with respect to an unobserved sensitive attribute that might influence their lexical choices, or the ways medical professionals interact with them, may be closer in embedding space than warranted given their other (potentially partially observed) outcome-relevant characteristics.

Additionally, the empirical support for the authors' claims is limited (in keeping with the nature of the format)--- this is something that should be addressed in future iterations. One very minor suggestion is to use \citep when citations appear in-line to improve readability.

---

### Official Review · Reviewer_8dPR · 2026-02-21
**Review for Submission 37**

**Rating:** 3
**Confidence:** 3

**Summary:**

The paper introduces a new fairness definition called similarity group, which is a synthesis of group and individual fairness notions that enable the auditing and safeguards of fairness without relying on demographic metadata.

**Strengths:**

The definition of similarity group is novel and strongly motivated.

**Weaknesses:**

There are a few places where additional explanation would improve clarity:

1. It is unclear why all fairness definitions appear to drop to 0 when the number of groups is 7, and then increase again when the number of groups is 8. This behavior is surprising and should be explained.

2. How is $\phi$ defined in the example?

3. It is unclear how Eq. (1) and Eq. (2) are related (e.g., whether one is derived from the other, or whether they represent different parts of the framework). Please clarify this connection.

### Minor comments

The citation style appears to be inconsistent / not in the correct format.

---

### Meta-Review · Area_Chair_4GFP · 2026-02-24

**Recommendation:** Reject
**Confidence:** 4

**Metareview:**

Two reviewers have evaluated the paper with mixed conclusions (summary assessment: 2x borderline).

The reviews note the novel contribution/ idea that allows to group individuals and perform fairness assessments in the absence of access to demographic data. The research is characterized as well motivated and the approach apparently supported by early empirical demonstrations.

However, the reviews also raise questions with respect to the robustness of the proposed procedure to methodological choices and possible latent confounding. The experiments also include curious results that are not well explained.

More broadly, I agree that the work is generally interesting, but it also seems like the topic of the paper is too complex to be fully addressed with such limited space. I therefore motivate the authors to continue this line of work, but propose the paper to be rejected at this stage.

---

### Decision · Program_Chairs · 2026-03-02

Reject